# An Integrated Framework for Image Acquisition, Processing, and Analysis Procedures for Automated Damage Evaluation of Concrete Surfaces

**DOI:** 10.3390/ma17040813

**Published:** 2024-02-08

**Authors:** Haixu Zhang, Cassandra Trottier, Leandro F. M. Sanchez, Anthony Allard

**Affiliations:** 1Department of Civil Engineering, Faculty of Engineering, University of Ottawa, Ottawa, ON K1N 6N5, Canada; hzhan181@uottawa.ca (H.Z.); leandro.sanchez@uottawa.ca (L.F.M.S.); 2Expertise and Material Engineering, Englobe Corporation, Québec, QC G1P 4S9, Canada; anthony.allard@englobecorp.com

**Keywords:** visual inspection, condition assessment, artificial intelligence, alkali-silica reaction, concrete surface crack, machine learning, cracking index, total crack length, image analysis, image processing

## Abstract

Concrete surface cracks serve as early indicators of potential structural threats. Visual inspection, a commonly used and versatile concrete condition assessment technique, is employed to assess concrete degradation by observing signs of damage on the surface level. However, the method tends to be qualitative and needs to be more comprehensive in providing accurate information regarding the extent of damage and its evolution, notwithstanding its time-consuming and environment-sensitive nature. As such, the integration of image analysis techniques with artificial intelligence (AI) has been increasingly proven efficient as a tool to capture damage signs on concrete surfaces. However, to improve the performance of automated crack detection, it is imperative to intensively train a machine learning model, and questions remain regarding the required image quality and image collection methodology needed to ensure the model’s accuracy and reliability in damage quantitative analysis. This study aims to establish a procedure for image acquisition and processing through the application of an image-based measurement approach to explore the capabilities of concrete surface damage diagnosis. Digitizing crack intensity measurements were found to be feasible; however, larger datasets are required. Due to the anisotropic behavior of the damage, the model’s ability to capture crack directionality was developed, presenting no statistically significant differences between the observed and predicted values used in this study with correlation coefficients of 0.79 and 0.82.

## 1. Introduction

Among the various types of internal swelling reactions (ISR) that may affect concrete’s performance and serviceability, the alkali–silica reaction (ASR) is the most reported type in Canada [1,2,3]. The ASR initiates from the chemical reaction between the alkali hydroxides (i.e., Na^+^, K^+^, and OH^−^) from the pore solution and some unstable mineral phases (i.e., reactive silica) within the fine/coarse aggregates, generating a gel-like secondary product that swells upon moisture uptake, leading to expansive pressure within the concrete beyond the tensile strength, thus cracking the concrete. Cracks begin within the concrete and develop into macro-cracks on its surface (Figure 1) [3]. ASR-related damage is distinguished by its map cracking pattern [4] that frequently appears on concrete surfaces and can be readily visible, serving as an early indicator of potential damage in concrete structures. Monitoring cracks over time provides critical insights into the progression of damage, as the presence of the ASR can result in harmful distress to concrete mechanical properties, stiffness, serviceability, and durability [5]. However, assessing the presence and severity of cracks remains an initial step in evaluating the overall condition of the concrete structure [3]. Currently, the evaluation of concrete elements affected by the ASR starts with the visual inspection process, aimed at qualitatively examining signs of deterioration on the surface.

To perform a quantitative analysis of the damage extent to monitor volumetric expansion over time, researchers have introduced a surface crack mapping technique known as the cracking index (CI). This non-destructive quantitative tool assesses the degree of damage, estimating concrete expansion by measuring the widths of cracks observed on the ASR-affected concrete surface [6,7]. However, the CI technique can be subjective and sensitive to the inspector’s experience, resulting in less accurate and reliable assessments [8], notwithstanding its time-consuming and environment-sensitive nature (i.e., variability due to time of day, sunlight position, time of year, weather, overcast), among other challenges thoroughly described in [9]. Moreover, the method varies internationally; in France, where delayed ettringite formation (DEF) is one of the leading causes of premature damage, the CI includes measurements along its diagonal [10]. Nevertheless, this indicates the method’s transferability to other mechanisms and ability to be easily modified digitally due to the similarities in procedures and crack patterns [11,12].

As such, image analysis techniques integrated with artificial intelligence (AI) have proven increasingly efficient as tools to capture patterns on concrete surfaces. However, many applications have been limited to qualitative assessments, such as crack identification without association to ASR-related damage or conducting a quantitative analysis of the observed damage in general [13,14]. Some research on automated crack detection models is mainly trained by datasets comprising a sufficient number of small-sized images (e.g., 336 × 339 pixels, 1 cm × 1 cm) containing partial details of cracks [15,16,17]. Although the CI was developed as a practical tool decades ago, new technologies and enhanced resolution obtained with conventional digital cameras allow the development of an automated procedure to capture the extent of ASR-related damage at the surface of a concrete element. The CI aims to quantify damage by summarizing crack measurements over a specific area (e.g., total analyzed area of 0.5 m × 0.5 m) [6]. Nevertheless, there is a shortage of datasets containing the required images to train machine learning models for quantifying the extent of cracking (i.e., minimal annotated datasets) [16,17]. Moreover, most of the literature is focused on the crack detection task [18,19,20,21,22,23,24,25,26,27,28,29,30,31,32] where images are taken near the surface and without quantification of the damage. In a condition assessment, a quantitative value is required to inform the decision regarding the next steps and to monitor the increase in the damage over time to capture the rate of the damage. Therefore, a rigorous protocol is essential to transform images into quantifiable features to advance recognition and damage quantification models.

## 2. Scope of This Work

Annotated and quantifiable images for use as datasets presenting damage extents captured by cracking intensity are lacking. This study aims to develop a flexible image acquisition and processing protocol to accommodate varying conditions (i.e., digital camera type and size of the analyzed area) for reliable quantitative evaluation of cracked concrete surfaces through image analysis. This work can therefore serve as a precursor to training machine learning models to detect crack patterns with quantified intensities that include an established image acquisition procedure, ensuring standardized images and data quality.

## 3. Materials and Methods

### 3.1. Concrete Specimen Employed for Image Acquisition

The database of this project refers to annotated images capturing crack size at the concrete surface along with crack tracing to measure both the quantity and length of cracks. To ensure reliability and quality, images will be analyzed under different light conditions using different crack-quantifying approaches, and comparative assessments will be performed between manual and digital evaluations. The data must reflect known causes and extents of damage; therefore, images were taken from laboratory-made concrete blocks affected by ASR-induced damage within a controlled environment. These blocks have reached their optimum expansion level, as detailed in the study by [33]. In this study, seven concrete blocks (450 mm by 450 mm by 675 mm) were used as image and data acquisition objects. These blocks had been manufactured in a laboratory environment and had previously been the subject of investigation in studies [8]. The concrete mixtures of the blocks included highly reactive coarse (i.e., Springhill) and fine (i.e., Texas) aggregates. The blocks also featured various reinforcement configurations, which are detailed in the previous study [34] and shown in Table 1. The concrete blocks were produced following the ASTM C1293 [35] mixture proportions and stored under conditions of 38 °C and 100% relative humidity to expedite the development of ASR. The blocks were kept in the same conditions until they reached their ultimate expansion, at which point they were removed for use in this study. These blocks were selected for this study due to their controlled fabrication and exposure conditions (to minimize variables, such as sunlight and temperature), known level of damage (measured through expansion since the beginning of casting by [34]), abundance of cracking at the surface, and surface characteristics (flat, clean, free of defects, etc.) [6].

### 3.2. Primary Damage Quantification Assessment Method—Cracking Index (CI)

The cracking index (CI) is a technique for mapping cracks that quantitatively evaluates the degree of surface cracking in concrete elements. This method includes measuring and summing the widths of cracks along lines that are drawn as square boundaries on the concrete surface under examination [4].

The measured crack widths are then recorded and added together along each line, resulting in a cumulative crack width value for that specific line. The calculation for the CI is expressed in Equation (1), which was proposed by [4]:(1)$$CI=\frac{\sum \mathrm{Crack}\mathrm{openings}}{\mathrm{Base}\mathrm{length}}$$
where the crack opening is the sum of all crack width measured along the grid line in millimeters, and the base length is the length of the grid line with units in meters. Moreover, the inferred expansion estimation has been adapted for reinforced concrete by [33] and is described in Equation (2):(2)$$\varepsilon \left(\%\right)=\frac{CI}{n}$$
where ε is the inferred expansion calculated, CI is calculated based on field measurements, and n is the number of cracks encountered in the direction of interest.

To perform the CI measurement, a square frame with four lines is created on the severely damaged concrete surface. Typically, these lines are drawn parallel and perpendicular to the main restraint(s). A square of 0.5 m by 0.5 m is typically drawn on structure elements, and the size of the region of interest (ROI) frame can be adjusted for the smaller damage area. As such, the spacing between the lines may vary depending on the size and complexity of the concrete element. To obtain a reliable assessment of cracking using the CI method, it is recommended to create multiple CI reference grids on the surface of the most severely cracked structural components [6]. These components are typically the ones exposed to moisture and harsh environmental conditions, as well as areas where the ASR is expected to have occurred to a significant degree.

As shown in Figure 2a, each CI reference grid should contain a measurable amount of cracking. These grids serve as visual reference points for quantifying and documenting the extent of cracking in the concrete structure. To measure the width of cracks on the surface of the investigated element, a magnifying lens with a plastic crack comparator card is required for precision. Each line drawn on the surface should be divided by 10 intervals as reference points and to tabulate the crack amount and width in 10 steps for each side. Figure 2a illustrates a 25 cm by 25 cm 3D-printed reference frame used in this study since the full grid size exceeded the block surface. Figure 2b illustrates the ROIs where the concrete block surfaces were divided into 6 segments, and some overlap is noted due to the limited size of the surface.

### 3.3. Image Acquisition

Image acquisition is a crucial step for the database collection in this work while ensuring image quality. This section undertakes a comprehensive approach to the image acquisition procedure, wherein the images selected for analysis must exhibit clear feature characteristics based on the goal of the analysis. In this study, cracks caused by ASR are the desired feature and are more specifically used to quantify their characteristics, such as orientation, length, and width, to highlight their distribution and pattern. The image acquisition configuration used in this study is described below.

Images are expected to demonstrate the features (i.e., cracks), differentiated pixel-wise, which allows the edge of each crack to be segmented from the surrounding background either manually or automatically in subsequent image processing. The following equipment and software were employed in the image acquisition process of this study:Camera: A digital single-lens reflex (DSLR-Canon EOS-T8i EF-S 18–55 mm f/4–5.6 IS STM, Brampton, ON, Canada) camera was used, capturing images at a resolution of 24.1 megapixels, maximum image size of 6000 × 4000 pixels, and International Organization Standard (ISO) sensitivity range of 100–25,600. The camera was equipped with a versatile lens with a focal length range of 18 mm to 55 mm and up to 1.6× zoom capability.Remote shooting software: Commercial software (EOS Utility 3.17.0 included with the camera) compatible with the camera facilitates the connection of the camera to a computer for remote shooting and camera control using a USB cable or Wi-Fi connection. A remote shooting option is preferred during the capturing process since it minimizes vibrations. The connection also enables real-time review of captured images on the computer screen, serving as the first stage of the quality validation.Lighting: Compact and portable LED light kits (containing 480 LED units, 3360 Lux/m, color rendering index of ≥96) were used to enhance the visibility of crack features, with adjustable intensity and color temperature (i.e., 3200 K—yellow/amber) to 5600 K—white/cold). The adjustments are modifiable from the unit itself, which changes the white balance of the light.Reference frames: A 3D printed reference frame (25 cm × 25 cm) and a 3:2 ratio frame were used for image processing and perspective distortion correction. The frames were placed on top of the interest area and ensured a consistent camera-to-surface distance (i.e., not varying zoom ranges and adjusting the reference frame to the picture frame size), minimizing parameters that could affect image quality.

Each side of the 3D printed square was divided into 10 equally spaced intervals normally used to record the CI data and used for scaling purposes. The image acquisition procedure involved setting up the equipment (Figure 3) and capturing images under different lighting conditions: white (cold, 5600 K), maximum yellow (warm 3200 K), and white/yellow (daylight, 4800 K). Camera settings and positioning for each image captured were kept consistent.

It is worth mentioning that the concrete blocks were lifted onto a table to allow for a comfortable working height, thus minimizing human error. The lights were placed on either side of the camera, and the camera was brought close to the surface of the concrete block until only the reference frame was brought into the field of view. The procedure will ensure that each image captured by the camera captures the ROI with the maximum quality, where the minimum crack width as per the comparator card is visible in the images. Furthermore, many images may exhibit perspective distortion, which occurs when the camera’s imaging frame is not parallel to the target plane during photography, leading to a trapezoidal shape instead of maintaining the intended square shape, as depicted in Figure 4. To address perspective distortion, a 3:2 frame was used to aid in adjusting the camera’s shooting angle and position. Detailed instructions on how to address the perspective are further illustrated in the following step-by-step procedure for image acquisition.

Frame attachment: The 3D-printed CI frame and the 3:2 perspective correction frame were attached to the damaged surface. The purpose of the outer frame was to locate the ROI, and the inner frame is used to aid the camera to mitigate perspective distortion.Camera setup: A tripod was used to minimize movements and ensure that the camera is aligned with the center of the segment to prevent tilted or skewed images. The camera was set to manual focus mode, and the lens zoom was adjusted to the minimum level (1.0×). The camera’s position was adjusted to ensure the outer frame overlaps with the camera view’s border. When multiple ROIs need to be collected on the same surface, the distance between the camera lens and the objective surface was measured for further reference.Lighting setup: LED lights were placed at the same height as the camera and at 45-degree angles relative to the subject. These LED lights offered the capability to modify the color temperature within the range of 3200 K to 5600 K, allowing for a transition from a warm yellowish color to a colder white tone.Remote shooting: Connecting the remote shooting software on the computer to the camera allowed for full control of the target feature to be captured.Perspective adjustment: The camera lens’ zoom ring was adjusted to view the perspective frame (as shown in Figure 5). The height and shooting angle were tuned to guarantee that all four corners of the frame fit within the camera’s view. After the adjustment, the camera’s shooting angle should be corrected to be perpendicular to the targeted surface.

6.Enhance image quality: The zoom ring was adjusted to the maximum level (i.e., fully zoomed-in) until a crack or any other surface feature is in focus. This ensures the best image quality. The zoom ring was then brought back to its original position, and the image was captured. The light conditions were then modified, and a new image was captured.

### 3.4. Image Processing

Images require further processing for quantified data collection purposes. Other than perspective distortion, images captured using a DSLR camera can have image distortion due to the camera’s wide angle lens structure, which arises from aberrations near the image edges. This type of problem has been introduced by Stankiewicz et al. (2018), where lens distortion represents a departure from the theoretical projection outlined in the pinhole camera model [36]. This phenomenon constitutes an optical aberration, causing straight lines within the scene to appear curved or distorted within the resulting image (Figure 6). This distortion is generated as a bending or curving effect along the edges of the reference frame within the image, which may induce inaccuracies when collecting quantitative data during subsequent post-processing.

To remove image distortion caused by a camera lens, two methods are available: (1) manually correcting the barrel distortion using an open source or commercial image processing software or (2) automating the procedure using a camera calibration algorithm, such as the OpenCV platform [16]. Zhang et al. (2000) developed a camera calibration algorithm and illustrate the camera calibration process using a mathematic model based on the OpenCV platform [37]. The pin-hole camera model establishes a mathematical relationship between the three-dimensional (3D) coordinates in the real world and their projection onto a two-dimensional (2D) image plane. The calibration process typically involves capturing a set of calibration images, estimating camera parameters, and applying distortion correction to subsequent images. The following is a general overview of the steps involved:Capture calibration images: A series of images (around 20–30) of a calibration pattern were taken from different angles and distances within the camera’s field of view, ensuring that the calibration pattern covered a significant portion of each image. The study from Zhang et al. (2000) indicated that the chessboard/checkerboard pattern is commonly used as a calibration target, which includes a grid that has distinct corner geometry allowing for precise calibration calculations [37]. In this study, an aluminum board was used for rigidity and durability, and the pattern was printed by a local photography shop.Detect calibration pattern corners: Detecting the corners of the calibration pattern was accomplished by inputting the calibration image set into a Python code named “calibration” [37] executed within the PyCharm 2023.2.5 software, an integrated development environment (IDE). This code can identify the corners of the calibration pattern in each image by detecting substantial changes in color in a pixel-wise manner.Calculate camera parameters: Calculating camera parameters involves utilizing the calibration algorithm to estimate intrinsic camera parameters (such as focal length, distortion coefficients, and principal points) as well as extrinsic parameters (comprising rotation and translation vectors). These estimations are derived from the identified corners within the calibration images.Undistort images: Another code named “undistortion” [37] is employed to fix the image barrel distortion within the assigned folder with the camera parameters (intrinsic and extrinsic matrix) input calculated using the “calibration” code from step 2. This code processes all subsequent images, utilizing the intrinsic camera parameters as input. This operation entails applying inverse distortion equations to each pixel, effectively eliminating the distortion effect.

Once the distortion parameters are applied to the images, the previously displaced pixels within the image will be restored to their original positions. The precision of the “undistortion” process is influenced by the quantity of calibration images captured with the specific camera. The camera matrix and distortion coefficients obtained from the calibration can be stored using write functions in NumPy [38], allowing them to be readily accessed for future applications.

After addressing the distortion issue, the subsequent step involves cropping the images to fit within the confines of the reference frame, ensuring they adopt a square shape, and resizing them to a uniform resolution of 3000 × 3000 pixels. The scale variation was managed by resizing the images in alignment with the reference frame and served as a practical approach since the distance within the reference frame is known (i.e., 25 cm).

To produce the quantification result, this study relies on an image analysis software [39]. The crack widths were measured digitally along the cropped images, and a new parameter referred to as total crack length (TCL) was employed. The TCL is determined by dividing the total crack length by the surface area, and it is a measurement that cannot be feasibly performed manually on-site.

In this study, a series of images featuring cracks are manually annotated using an image processing software (provided compatibility with the device used during the annotations) and a tablet with pen by tracing the cracks using the pencil function with a known pixel width (Figure 7). The annotation setup was selected due to availability; however, the process is not limited to this type of setup.

The mask layer was then extracted, converted to an 8-bit image, and analyzed to determine the total number of traced pixels using the Particle Analyze function.

The results exported contain the crack lengths to which the calculation for the TCL can be applied (i.e., total crack length over analyzed area). The results are not limited to lengths but can also describe preferential crack orientation. Figure 8 summarizes the essential steps used in this study. Nevertheless, this study’s methodology aims to enhance the accuracy of crack assessment through image analysis as opposed to manual measurement.

## 4. Results

### 4.1. ASR Damage Measured Using the Cracking Index (CI)

The cracking index (CI) outcomes for seven concrete blocks were manually recorded by two operators and calculated as per Equation (1). Figure 9 illustrates the measured expansion as a function of CI, which exhibits an overlap between different types of blocks, yet some clustering is observed. The calculated expansion (Equation (2)) of 2D-TX blocks (reinforced in two directions and made with reactive Texas sand) can spread between 0.25% and 0.86%, and the CI can range from 0.35 mm/m to 14.50 mm/m. Furthermore, 2D-SP blocks (reinforced in two directions and made with reactive Springhill coarse aggregate) have a narrower range of expansion compared to 2D-TX blocks, but the CI can vary from 5.25 mm/m to 12.56 mm/m when the expansion level is around 0.65%. In contrast, the unreinforced type of block has a more concentrated CI value (2.9 mm/m to 5.9 mm/m) and expansion level (0.23% to 0.50%). Moreover, the 1D-SP blocks (reinforced in one direction and made with reactive Springhill coarse aggregate) show the narrowest ranges between 2 and 5 mm/m for expansions of 0.25–0.33%. Overall, a good correlation was found at a R^2^ value of 0.72.

### 4.2. Operator Sensitivity to Damage Quantification

Two operators participated in the manual CI measurement to quantify the variability between them. To assess the statistical significance and relevance of the results obtained from various factors, paired *t*-test results of CI measurements of 66 square segments by 2 operators are presented in Table 2. A strong positive correlation with a Pearson correlation coefficient of 0.95 was observed between the two variables. The average CI values obtained from the two operators were 5.74 and 5.88 with variances of 7.51 and 7.08, respectively. The calculated *p*-value was 0.37, which exceeds the significance level of 0.05. As a result, the null hypothesis cannot be rejected, suggesting that there is no statistically significant difference between the pairs of CI measurements manually measured by 2 operators.

### 4.3. Effect of Light Temperature on the Image Analysis Result

A comparative analysis of CI and the novel measurement of total crack length (TCL—length over area, mm/cm^2^) results was conducted over 72 digitized images under three distinct lighting conditions, including warm light (yellow—3200 K), daylight (white/yellow—4800 K), and cold light (white—5600 K), as depicted in Figure 10. Four out of the 7 blocks were selected for this portion of the study, representing various components with different reinforcement settings: unconfined, reinforced in one direction (1D-SP), reinforced in two directions with reactive coarse Springhill aggregate (2D-SP), or reactive Texas fine aggregate (2D-TX). A similarity in the results can be observed for each light condition, where overlapping results are observed, yet the overall trend is conserved. The highest CI value is captured with daylight (white/yellow) at 12.4 mm/m. The corresponding CI value for cold (white) light is 10.9 mm/m, and that for warm (yellow) light is 8.8 mm/m Figure 10a). Meanwhile, the largest difference can be observed with a CI value of 3.6 mm/m. As for the TCL, similar trends are observed, where the results are comparable to CI yet smaller variations are obtained from the TCL (Figure 10b).

Due to the varied extent of damage in the concrete blocks, the CI results range from 3.9 mm/m to 12.4 mm/m, reflecting some degree of variance within the group. However, ANOVA analysis (Table 3a) across the four different types of blocks under various lighting conditions indicates that varying lighting conditions does not significantly affect the CI results obtained from image analysis. Statistically, Table 3 presents a *p*-value exceeding the significance threshold of 0.05, indicating insufficient evidence to reject the null hypothesis (i.e., no significant difference in results among different lighting conditions). This conclusion is further supported by the fact that none of the “F values” exceed its corresponding “F-critical” value. Similar findings apply to the TCL image analysis results (Table 3b). TCL values range from 2.6 mm/cm^2^ to 4.7 mm/cm^2^. ANOVA analysis reveals a *p*-value higher than the significance level of 0.05, and the tabulated F value is below the F-critical value, reinforcing the absence of significant differences among the three sets of light condition results.

Based on the comparison, it can be concluded that the light conditions have an influence on the results, but the effect is minimal and can be considered negligible.

### 4.4. Digital Cracking Index

The digital image-based analysis is a different approach to collecting the measurement of cracks using pixel-wise annotations on images. Figure 11 illustrates the digital CI results from four selected blocks (one from each configuration as previously mentioned). Compared with the manual CI results, digital CI represents a more distinctive CI and expansion level for different types of blocks. CI values for unreinforced blocks ranged from 0.25 mm/m to 2.50 mm/m, with a calculated expansion level ranging from 0.22% to 0.55%. Both types of blocks reinforced in 2 directions (2D) exhibit more severe damage, and their expansion level similarly ranges from 0.50% to 0.78%. Interestingly, the digital CI results for the two types of 2D blocks show a significant difference (ranging from 6.3 mm/m to 11 mm/m for 2D-SP blocks and from 9.38 mm/m to 12.5 mm/m for 2D-TX blocks). Interestingly, more prominent clustering is observed using the digital CI when compared to the CI performed manually as observed in Figure 11a. Digitizing the CI was found to be effective as observed in Figure 11b where datapoints cluster along the 1:1 line.

### 4.5. Total Crack Length (TCL)

The total crack length (TCL) was measured through image analysis and calculated using a unit in mm/cm^2^. Figure 12 illustrates the calculated expansion from the digitally recorded crack widths as a function of the TCL over 24 segments under artificial light in daylight conditions (4800 K). The data points exhibit a broad distribution while generally clustering into distinct groups. The TCL values within all types of blocks range widely, ranging from 2.3 mm/cm^2^ to 4.6 mm/cm^2^. Notably, Block B5 (i.e., unreinforced, SP) displays the most modest expansion level (0.23–0.37%, computed based on CI values derived from image analysis) among the four types of blocks.

### 4.6. Crack Orientation

Through image analysis, the crack orientation can be obtained as an output that can be further illustrated using histograms to depict the most frequently observed crack directions. Figure 13 presents the histograms for a reinforced block and an unreinforced block. The distribution in the reinforced block histogram (Figure 13a) shows a preference for the 0-degree region (horizontal) and a lesser tendency towards the 90-degree region (vertical). This suggests that more cracks are oriented horizontally rather than vertically. In contrast, the unreinforced block exhibits an orientation angle that does not concentrate in any range of direction (Figure 13b). Table 4 presents the results for all blocks where the 2D reinforced blocks show average orientation angles of 35.37 and 33.12 degrees. These values are slightly less than that of the 1D reinforced block at 38.20 degrees followed by the unreinforced block at 45.46 degrees.

## 5. Discussion

### 5.1. Capturing Damage with Crack Directionality in Reinforced Concrete

The expansion calculated using Equation (2) from the data collected in this study was compared to measured expansion (from Table 1), and the data are presented in Table 5 where non-negligeable differences are observed from 23% to 65%. From Figure 9, a wide range of expansion values was obtained; however, some degree of clustering was observed, indicating that crack directionality due to reinforcements may influence the CI measurements. It is to be noted that although the CI calculates cracks in two directions, its final output is a one-dimensional value such that crack widths are measured along a line. In addition, in practice, the CI is performed in the center of a concrete surface to minimize the edge effect thus, influencing the representation of damage throughout the concrete. It is to be noted that in this study, images taken from the entire surfaces of the blocks were used to optimize the dataset therefore introducing such large differences.

Considering the apparent observable crack directionality, the CI was divided into two directions to capture horizontal cracking and vertical cracking. Figure 14 presents the average directional CI results from both operators plotted along a 1:1 line; this plot indicates that cracking is more significant in the horizontal direction. Note that the x-axis represents the crack CI measurement from cracks aligned in the horizontal direction, and vice versa.

By evaluating the data clusters where different types of blocks were grouped to assess the impact of reinforcement on crack patterns, it can be observed that all data points from blocks without reinforcement cluster closely around the 1:1 line, with the highest CI value of 4.20 mm/m for vertical cracking and 5.10 mm/m for horizontal cracking. This suggests that their horizontal CI values closely align with their vertical CI values since these blocks can expand in all directions without restraint. In contrast, all reinforced blocks exhibit CI values in horizontal cracks that are much greater than those in vertical cracks. The block is reinforced in one direction and has an average CI value of 2.43 mm/m for horizontal cracking and 0.81 mm/m for vertical cracking, which are lower than the value observed for blocks reinforced in 2 directions in both vertical and horizontal cracking directions. For blocks reinforced in 2 directions, 2D-SP has an average CI of 3.86 mm/m for horizontal cracking and 1.39 mm/m for vertical cracking, while 2D-TX has an even larger crack width open with values of 7.458 mm/m for horizontal and 1.439 mm/m for vertical cracking directions.

Figure 15 displays the expansion level calculated using Equation (2) against the CI value for horizontal and vertical cracking directions, separately. Based on 132 observations, it is evident that the calculated expansion levels as measured based on the horizontal cracking direction exhibit a larger mean range of expansion (0.2% to 1.6%) compared to the vertical cracking direction (0.2% to 0.8%), especially for reinforced concrete blocks. It is apparent that both CI and expansion values plotted for 1D-SP, 2D-TX, and 2D-SP types of blocks are notably higher than those for unreinforced blocks (Figure 15a). In contrast, when measuring cracks in the vertical direction (Figure 15b), the values tend to cluster within a certain range for each block. This observation reveals that the presence of reinforcement has no effect and leads to increased crack openings and expansion along the reinforced direction. Corroborating the existing literature [8,14,33], the expansion measurements performed on various surfaces of the reinforced blocks revealed that the propagation of induced expansion transfers from reinforced directions to directions with lower or no reinforcement and results in more expansion in the direction with the least amount of reinforcement (perpendicular to the main reinforcing bars). This observation suggests that ASR-induced expansion occurred to a greater degree in the direction where the reinforced blocks faced less or no restraint. For instance, 2D-SP blocks were formed by reactive coarse aggregate (Springhill-SP) and non-reactive natural sand, with reinforcement in the transverse and longitudinal direction. The average result for 2D-SP blocks exhibited an average horizontal CI of 4.05 mm/m compared to a vertical CI of 1.45 mm/m. This disparity indicates that the average expansion in the horizontal direction (0.38%) was constrained, leading to a more pronounced expansion in the vertical direction (0.83%) that experienced no restraints.

Clustering in Figure 15 shows that the calculated expansion as a function CI for horizontal cracking is more distinct when compared to that observed in Figure 9 (i.e., undivided CI). Likewise, through the cracking orientation analysis, preferential cracking directions are observed with respect to the reinforcements. Therefore, in cases of reinforced concrete presenting cracking directionality, quantifying the cracking orientation may help to refine models used to calculate the expansion from the CI.

### 5.2. Variability to Consider When Digitizing

It is expected that the highest degrees of variability will be observed during manual operations as more factors can influence the variability as opposed to measurements obtained using digitized images. This reasoning stems from the variables often encountered when performing manual operations on-site or in a laboratory in which human error is more prominent. Therefore, concerns about operator variability and the use of artificial light have often been raised due to the nature of the conducted on-site visual inspections. Through validation in this study, the CI obtained on-site (called “manual CI” in this study) was not deemed sensitive to operator variability when both operators work alongside each other under the same conditions. Operator variability remained at 0.2 mm/m, which is considered negligible. As such, the operators’ variability is lower than that of the range used to indicate the volumetric changes around 1.85 mm/m per year [34]. Meanwhile, it was previously stated that artificial light conditions had a negligible effect on the outcomes. However, when digitizing the CI, variability may be observed due to image resolution; thus, a set of *t*-test results were obtained (Table 6). The CI was further divided into two directions. The average value for manual CI is 3.55 mm/m, whereas that for digital CI is 3.90 mm/m. The difference between these averages yields a *p*-value of 0.051, which is just on the border of the significance level of 0.05. Similarly, a *p*-value of 0.051 was found in the vertical CI results from both measurement approaches. Since both *p*-values are very close to reaching statistical significance, it suggests that there is a small chance of observing a significant difference in all measurements. Obtaining more convincing results may require a larger sample size.

The conditions affecting the results during each operation are indeed different such that digital CI measurements can be influenced by image quality provided by the camera itself or the computer/tablet screen used to measure the cracks. It is expected that lower image quality will provide a higher variability, while a better image quality will provide less variability as the crack features will be more resemblant to what is observed by the eye. A parametric study using various image qualities would help to further define the ranges of uncertainty. Furthermore, this correlation is for the observed surfaces used in this study, which do not include lower degrees of damage or different types of damage other than ASR and represent laboratory-made and exposed concrete. Further images following the proposed standard protocol in this study are required to refine these correlations and the dataset. Nevertheless, the CI served as an efficient tool to measure the ability to transition from manual to digital measurements of cracking on the concrete surface.

### 5.3. Image Analysis for Damage Evaluation

A new metric was considered to quantify the intensity of cracking similar to the total crack length (TCL), which considers the length of cracks (in one dimension) over an area (2 dimensions). The TCL alone cannot measure crack directionality; therefore, it was compared against the vertical and horizontal CIs, as illustrated in Figure 16. By comparing the slope of the trendline, the TCL may not be able to represent a good correlation with either the vertical or horizontal CI value. As the TCL increases, the value of CI between the two directions shows a significant difference, indicating the TCL may not be capable of representing the damage extent in either direction. Moreover, the correlations show an R^2^ of 0.39 for horizontal cracking and 0.06 for vertical cracking, which is considered a poor correlation.

The orientation angle of cracks can therefore serve as potential indicators. As previously mentioned, the configuration of reinforcement in certain blocks may contribute to an increased prevalence of surface cracks in the horizontal direction. The results of crack orientation are displayed in Table 4 and describe the overall crack propagation direction, thus providing a potential solution to convert TCL to a directional CI by assigning coefficients according to the angle within a certain range corresponding to the direction. In this study, Equations (3) and (4) are defined through linear regression using the image analysis results obtained in this work.
(3)$${CI}_{v}=3.271a_{v}\times CD-0.8698$$
(4)$${CI}_{h}=1.7653a_{h}\times CD-0.3718$$
where $a_{v}$ and $a_{h}$ are the total percentages of crack orientation from 0° to 30° and 60° to 90°, respectively, representing cracks propagating in horizontal and vertical directions, respectively. The CD is the TCL in mm/cm^2^, and CI_v_ and CI_h_ are the predicted CI results for the vertical and horizontal directions, respectively.

The two plots presented in Figure 17 compare the predicted CI results vs. manual CI in both directions to better determine the regression performance visually. The performance of the regression model can be assessed through residual analysis, as depicted in Figure 18. The residual values range from −2.0 mm/m to +2.3 mm/m for the vertical direction and from −1.0 mm/m to +1.0 mm/m for the horizontal direction, and a trend of heteroscedasticity is not observed. Notably, the histograms of the residual values in both directions closely approximate a normal distribution. This suggests that the prediction model (Equations (3) and (4)) has good regression with ground truth values.

The CI, CD, and crack orientation angle represent three distinct parameters that do not exhibit a direct relationship. Nevertheless, when considering CD and crack orientation collectively, they can jointly characterize crack behavior in terms of distribution and quantity. Their combined analysis can reasonably describe the cracking intensity in their respective directions, akin to the role CI plays. Additional findings presented in Table 7 provide compelling evidence to assess the model’s predictive capability. Based on the results of paired *t*-tests, the predictive outcomes of CI closely align with manual CI results of horizontal cracking, yielding an average *p*-value of 0.96 and a correlation coefficient of 0.82. These results suggest that there is no significant difference between the predicted and observed values. Similarly, in the vertical direction, the *p*-value is 0.90 with a correlation coefficient of 0.79, supporting a similar conclusion of a positive result.

Through the analysis presented above, the combination of total crack length and crack orientation has proven ability to summarize damage extent results similar to the output of CI. Since ASR development involves anisotropic propagation in reinforced concrete samples, accurately extracting damage from randomly distributed crack patterns can be challenging. However, the model introduced in this work successfully integrates crack quantity and crack propagation direction across the entire area of interest to correlate with an already established metric, the CI, for evaluating crack damage in two directions across the damaged surface.

Further refinement of this model is however necessary, especially when incorporating images from real structures in the field exposed to harsh climates. In addition, increasing the dataset of images of ASR surface cracking with known levels of internal damage can provide a more accurate and reliable approximation of the expansion levels (%). Moreover, the CI is normally performed in the center of a concrete surface to reduce the edge effect where less representative cracking is observed.

## 6. Conclusions

Digitization of visual inspections is becoming the standard practice in many applications, such as assessing the cracking of concrete surfaces. Among the mechanisms that cause concrete to crack, the alkali–silica reaction (ASR) causes a distinct map cracking pattern in which its intensity is quantified to evaluate the overall level of deterioration and its rate. The cracking index (CI) was initially developed to measure this cracking intensity in a practical manner on the field. Since its development, the availability and accessibility of high-resolution digital cameras have transformed visual inspections such that the assessment could be performed on a computer as opposed to at the site, thus reducing some of the variables encountered through such operations.

Variability in the CI computation between two operators was evaluated and was found to be negligible. Among other variable parameters that could be controlled in this study, artificial lighting was the most significant factor. All other variables were kept constant, such as the distance between the camera lens and the block surface, the camera zoom set to its minimal value, the position in the laboratory, angles adjusted for perspective distortions, and the area of analysis. Using three different light settings (all set to their maximum intensity to further reduce variability), image analysis was performed using the digital cracking index (where crack widths were measured on images) and the total crack length (where cracks were annotated/traced on images). It was thus revealed that the lighting conditions did not significantly influence the outcomes. Furthermore, converting the cracking index into a digital technique displayed a strong correlation, thereby validating the digitization process.A thorough protocol for image acquisition, processing, and analysis was developed in this project to facilitate the creation of similar datasets. Datasets of quantifiable features of concrete cracking are non-existent and crucial for machine learning development. The application of the protocol can lend itself beyond cracks in concrete caused by the ASR. This protocol therefore can serve as an established method to produce standardized images, ensuring data quality for further model development.By evaluating the expansion as a function of the CI, a linear trend was observed; however, the distinction between the various reinforcement configurations was not apparent when compared to the use of digitally measured crack widths to calculate the CI and the total crack length. The division of the CI into two directions, horizontal and vertical, helped to understand the role of crack directionality in the CI prediction using image analysis based on the total crack length. This enabled the development of two models for which horizontal and vertical cracking are treated separately as a function of their proportions, thus outputting a CI in both directions.Research is currently ongoing to validate the methodology when applied to real structures in the field subject to Canadian climate conditions where the level of ASR-induced internal damage is already known. This will help to establish the correlations between internal damage and corresponding external cracking due to the ASR. Moreover, the current study focused exclusively on ASR cracking; thus, further work is necessary when combined mechanisms are involved, such as mechanical cracking, shrinkage, and other internal swelling reactions.

## Figures and Tables

**Figure 1 materials-17-00813-f001:**
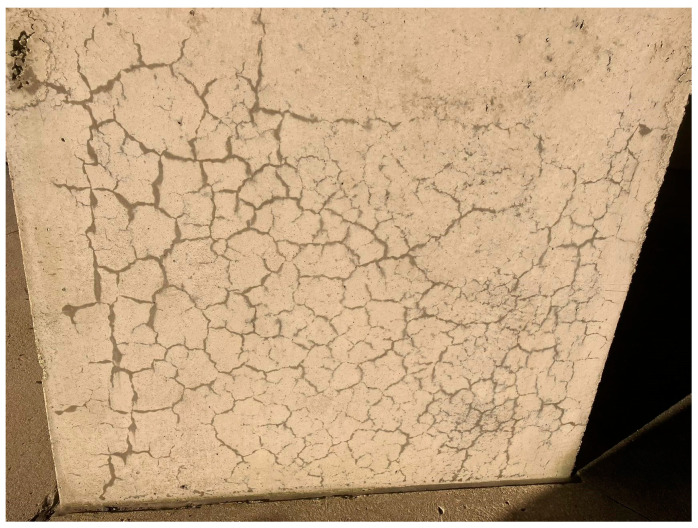
Concrete column affected by the alkali-silica reaction (ASR) presenting a map cracking pattern.

**Figure 2 materials-17-00813-f002:**
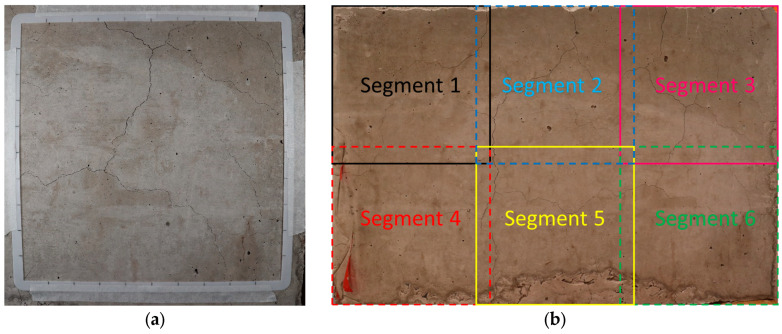
(**a**) A typical cracking index layout on a concrete surface using a 3D-printed reference grid representing one block surface segment and (**b**) a segmented concrete block surface.

**Figure 3 materials-17-00813-f003:**
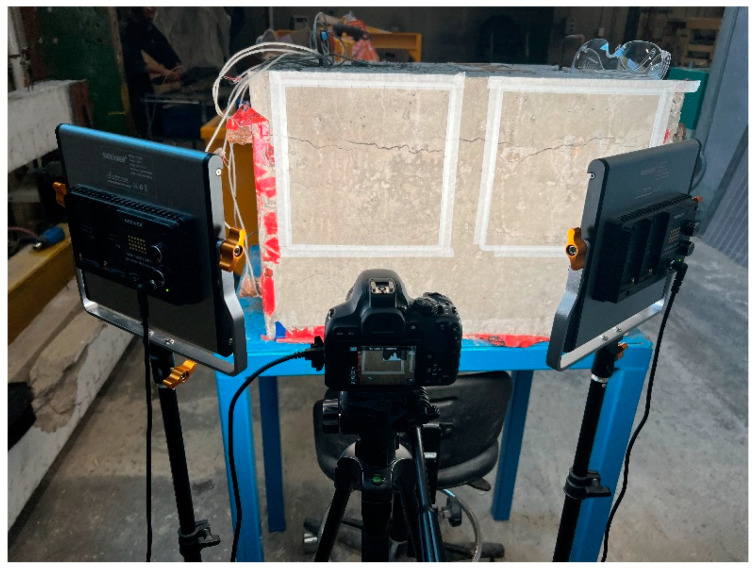
Image acquisition setup, including a digital single-lens reflex (DSLR) camera, LED light panels, and objective concrete elements with the reference frame attached.

**Figure 4 materials-17-00813-f004:**
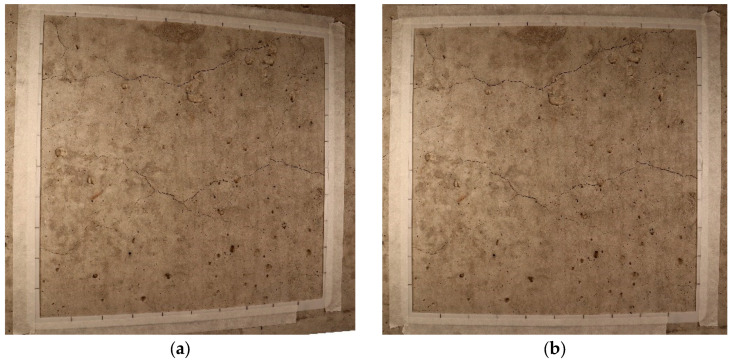
(**a**) Image with perspective distortion, (**b**) image after perspective correction. The image (**a**) has wider left side boarder than right side border, resulting in scale inconsistency for the whole image.

**Figure 5 materials-17-00813-f005:**
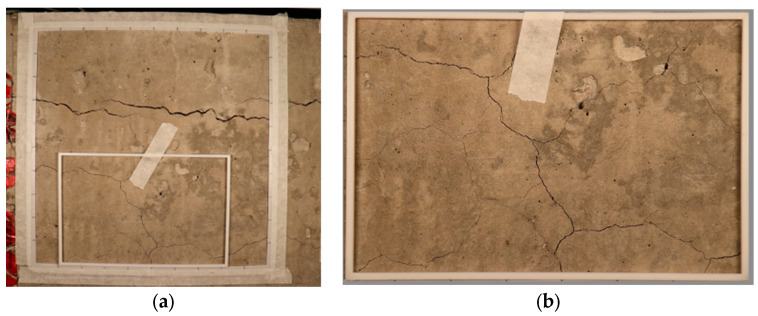
(**a**) A perspective frame affixed at the lower part of the reference frame and (**b**) the perspective frame in the full field of view after zooming in.

**Figure 6 materials-17-00813-f006:**
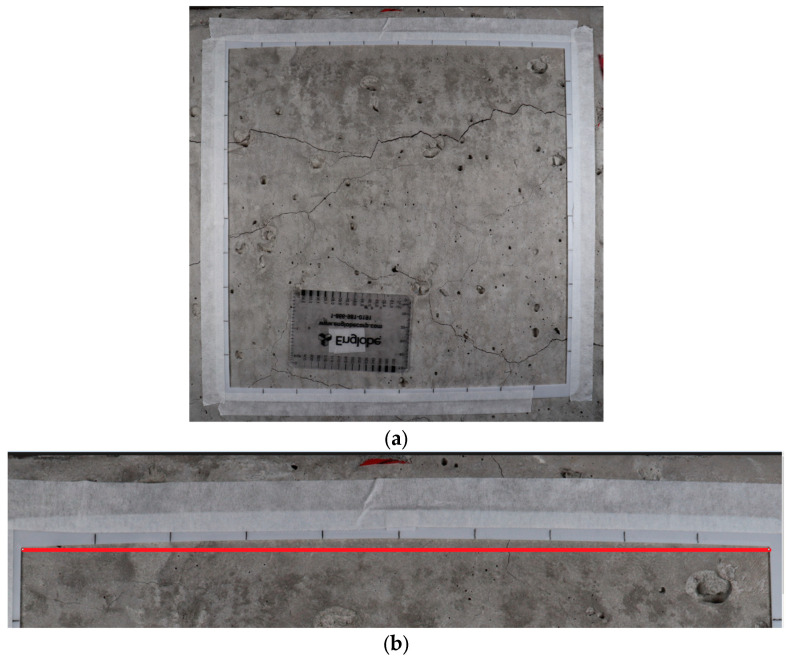
Barrel distortion that appears near the edge of the frame that caused the image to have outward distortion curves: (**a**) shows the full image and (**b**) shows the upper edge with the outward distortion curve showing deflection from a straight line drawn in red.

**Figure 7 materials-17-00813-f007:**
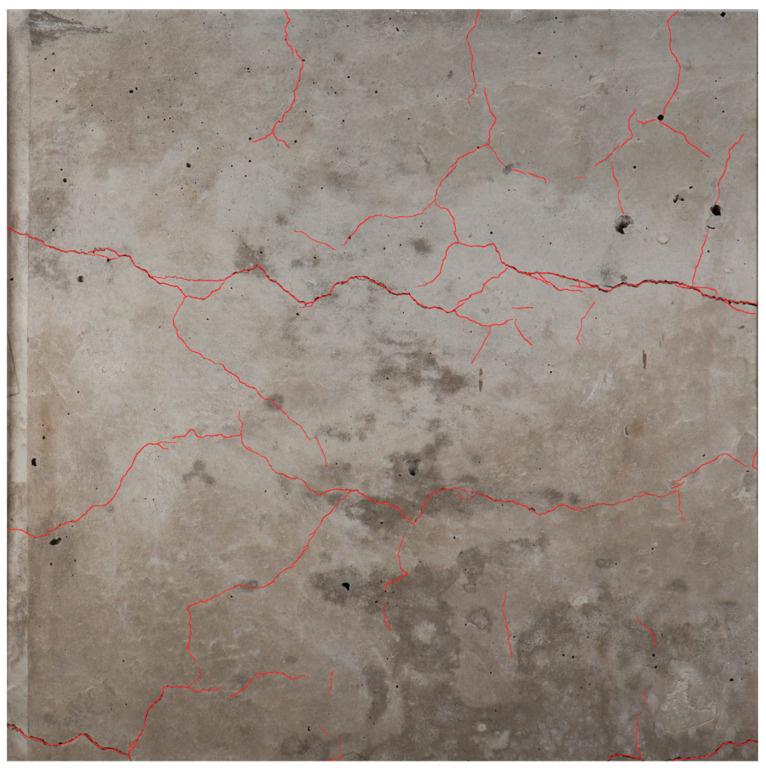
A traced image with its mask layer. The red lines represent the annotations that were manually traced.

**Figure 8 materials-17-00813-f008:**
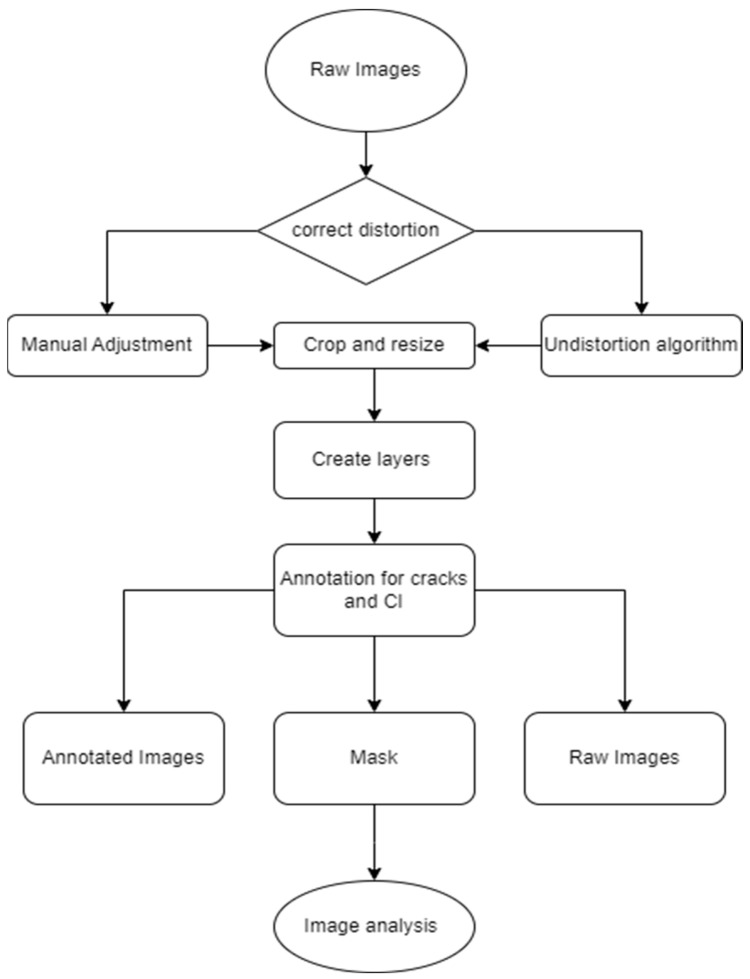
Image processing flowchart.

**Figure 9 materials-17-00813-f009:**
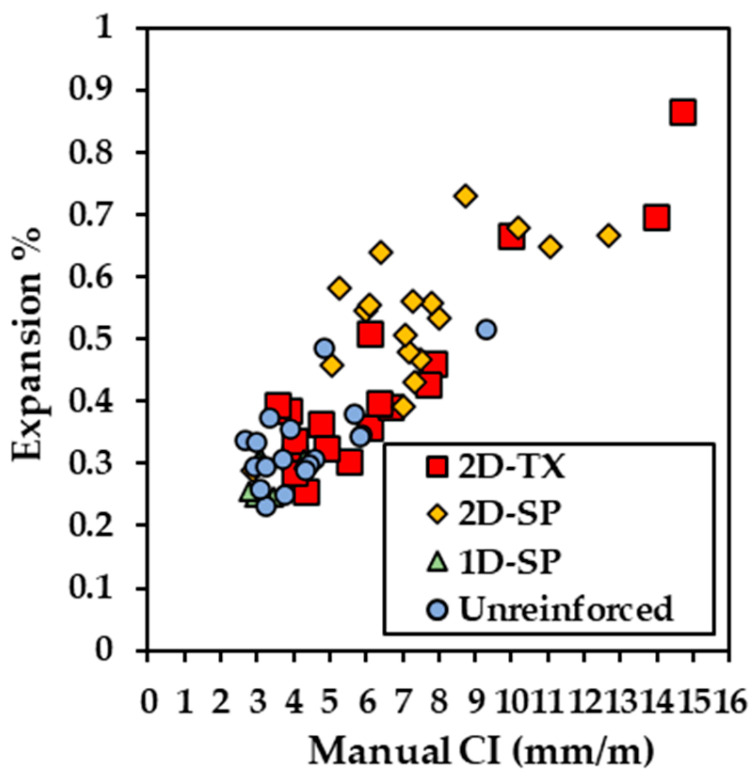
Calculated expansion level vs. average CI of two operators for 66 square segments.

**Figure 10 materials-17-00813-f010:**
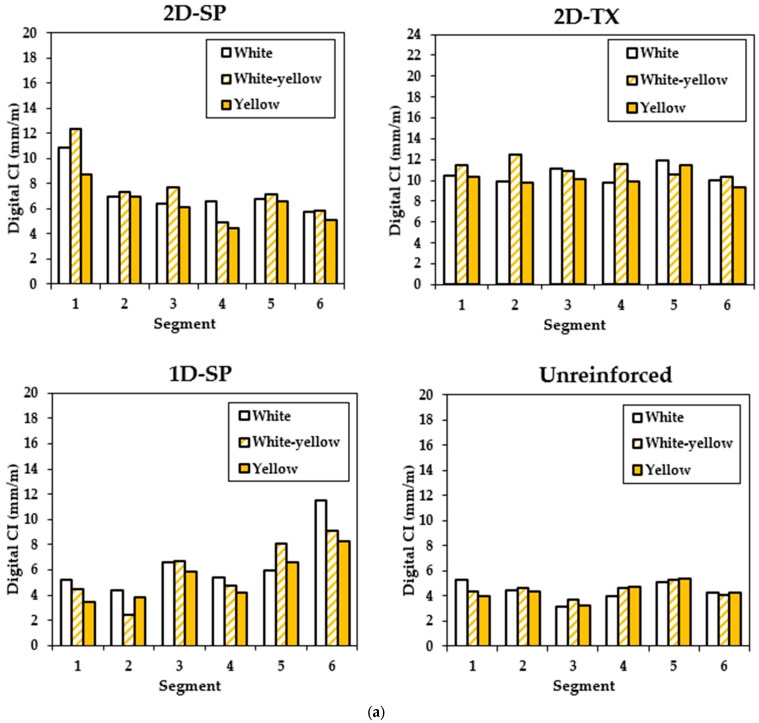
Comparison for 3 light conditions among 24 image segments from 4 types of blocks using (**a**) cracking index (mm/m) and (**b**) total crack length (mm/cm^2^). White bars = cold light (white—5600 K); striped white and yellow bars = daylight (white/yellow—4800 K); and solid yellow bars = warm light (yellow—3200 K).

**Figure 11 materials-17-00813-f011:**
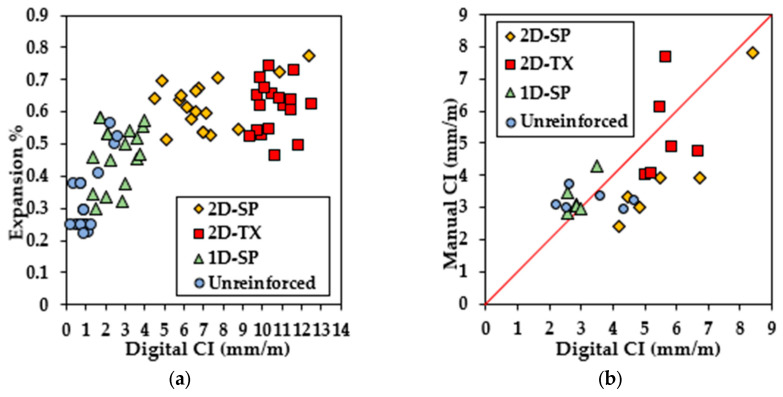
(**a**) Digital CI vs. calculated expansion level. (**b**) Manual CI vs. digital CI where datapoints are illustrated along the red 1:1 line.

**Figure 12 materials-17-00813-f012:**
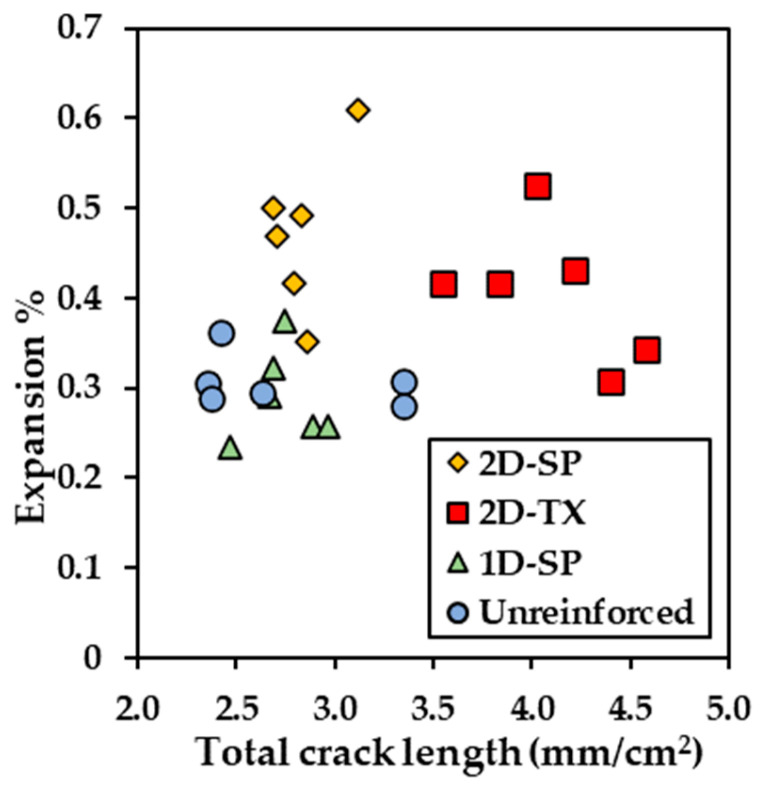
Calculated expansion level against the total crack length (TCL) value.

**Figure 13 materials-17-00813-f013:**
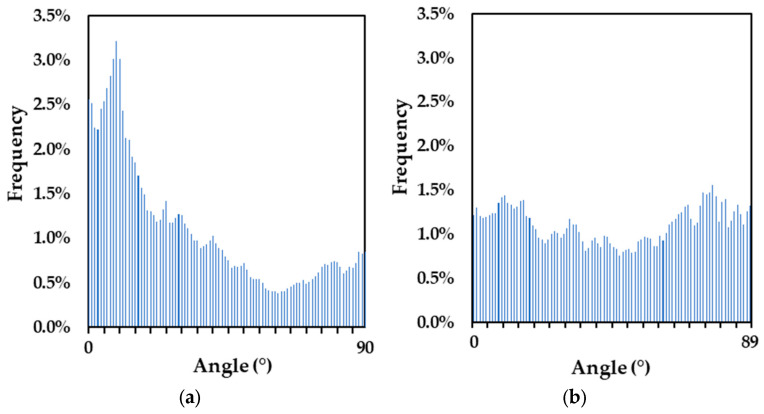
Histograms of crack orientation angle from one square segment of (**a**) a reinforced and (**b**) unreinforced block.

**Figure 14 materials-17-00813-f014:**
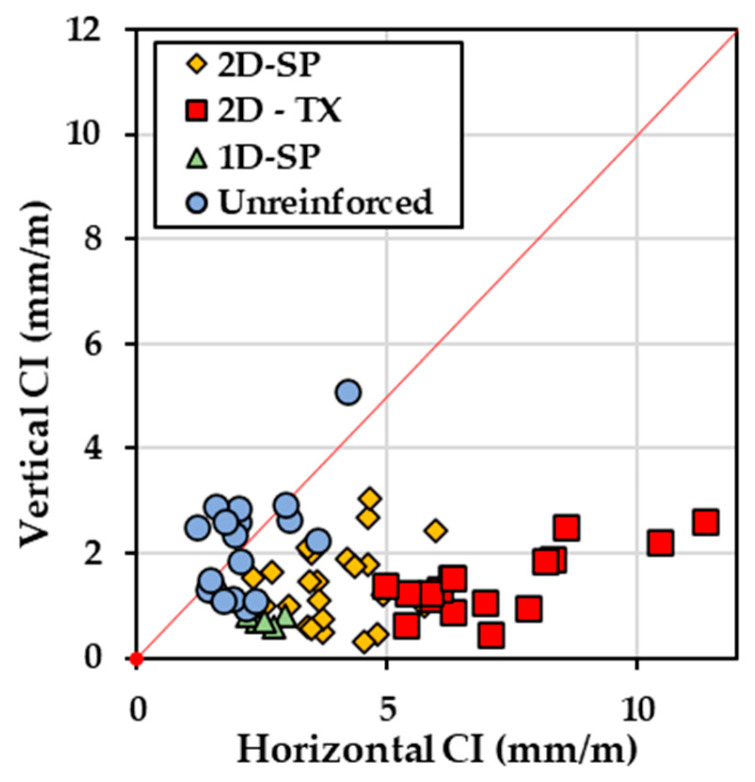
Plot of CI measurement collection for vertical vs. horizontal measurements in mm/m manually collected from 7 blocks with marker shape differentiated by concrete reinforcement configuration: (1) reinforced in two directions (2D) with reactive coarse (SP—blue square) or fine (TX—green square) aggregate, reinforced in one direction (1D) with SP coarse aggregate (triangle), and unreinforced with SP coarse aggregate (circle). Values are shown along the 1:1 red line.

**Figure 15 materials-17-00813-f015:**
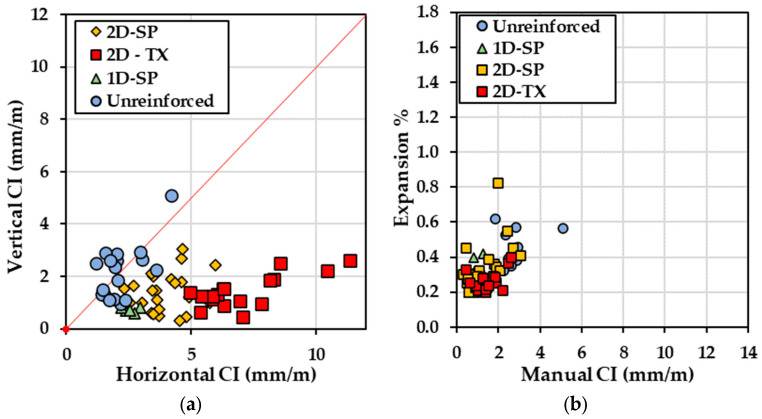
(**a**) Calculated expansion level vs. average CI result from manual measurements collected from 7 blocks with marker shape differentiated based on concrete reinforcement settings: circle—unreinforced, SP; triangle—1D-P; square—2D-SP; diamond—2D-TX. (**b**) Calculated expansion level vs. average CI result from manual measurements.

**Figure 16 materials-17-00813-f016:**
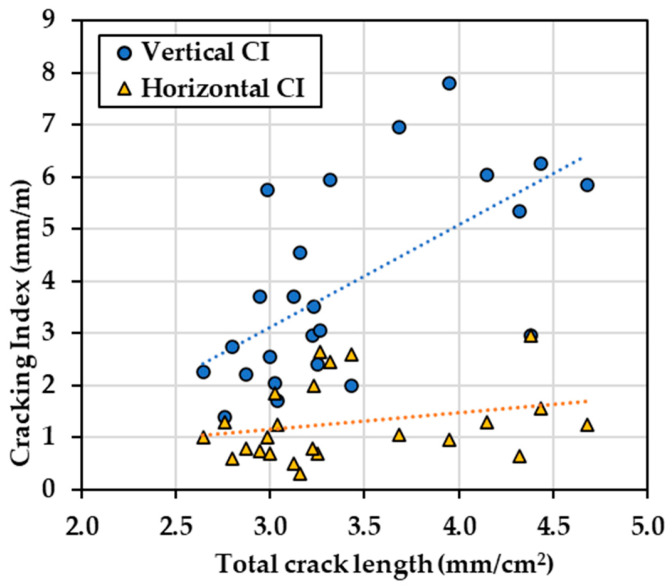
Cracking index (CI) as a function of total crack length (TCL). The dashed lines represent the trendlines.

**Figure 17 materials-17-00813-f017:**
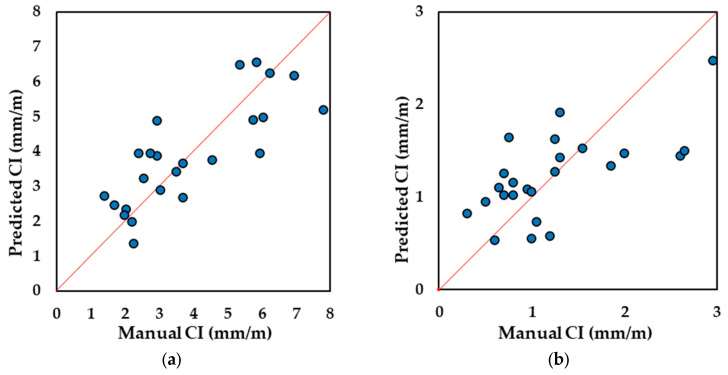
CI values predicted using the CD-orientation model in the (**a**) horizontal and (**b**) vertical directions.

**Figure 18 materials-17-00813-f018:**
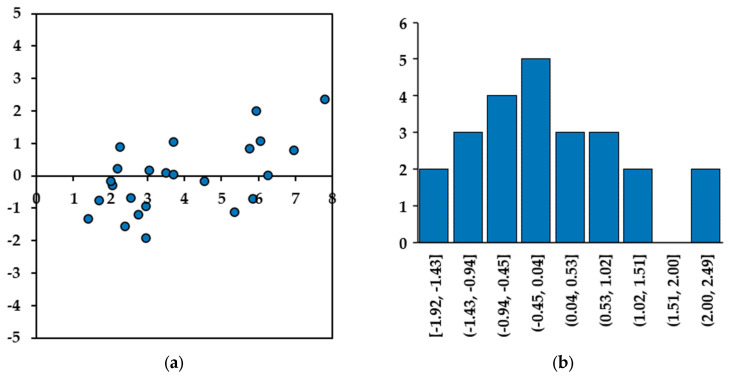
Residual analysis for CI prediction using CD and crack orientation: (**a**) horizontal residual plot, (**b**) horizontal residual histogram, (**c**) vertical residual plot, and (**d**) vertical residual histogram.

**Table 1 materials-17-00813-t001:** Description of selected concrete blocks.

Block Title	Description	Measured Expansion %	
		Longitudinal	Transverse
B3	Unconfined, SP	0.817	0.818
B5	Unconfined, SP	0.817	0.818
B12	1D-SP	0.817	0.818
B2D21	2D-SP	0.700	0.710
B2D19	2D-SP	0.700	0.710
B25	2D-TX	0.801	0.826
B2D29	2D-TX	0.801	0.826

Note: SP = Reactive Springhill coarse aggregate; TX = Reactive Texas sand; 1D = reinforcements in one direction; 2D = reinforcements in two directions.

**Table 2 materials-17-00813-t002:** Results from *t*-test comparisons between 2 operators’ manual CI measurements.

	Operator 1	Operator 2
Mean	5.74	5.88
Variance	7.51	7.08
Observations	66	66
Pearson Correlation	0.95	
Hypothesized Mean Difference	0	
df	65	
t Stat	−1.46	
*p* (T ≤ t) one-tail	0.37	
t Critical one-tail	1.66	

**Table 3 materials-17-00813-t003:** ANOVA results for (**a**) cracking index and (**b**) total crack length.

(**a**) **ANOVA of the Cracking Index under Different Light Conditions**			
**Block type**	**F**	** *p* ** **-Value**	**F-Critical**
2D-SP	0.54	0.59	3.68
2D-TX	1.86	0.19	3.68
1D-SP	0.51	0.61	3.68
Unreinforced	0.67	0.53	3.68
(**b**) **ANOVA of Total Crack Length under Different Light Conditions**			
**Block type**	**F**	** *p* ** **-Value**	**F-Critical**
2D-SP	1.95	0.18	3.68
2D-TX	0.12	0.89	3.68
1D-SP	0.46	0.64	3.68
Unreinforced	1.86	0.19	3.68

**Table 4 materials-17-00813-t004:** Mean value summary of average crack orientation angles from 5 concrete surfaces (unit in degrees).

Block Label	1	2	3	4	5	6	Average
1D-SP	36.99	35.91	30.38	44.74	43.35	37.83	38.20
2D-SP	33.32	40.23	25.17	34.82	43.70	35.00	35.37
2D-TX	40.22	26.09	34.34	33.93	30.34	33.77	33.12
Unreinforced	44.60	44.51	56.42	38.39	42.82	45.99	45.46

**Table 5 materials-17-00813-t005:** Comparison between measured and calculated expansions (average).

Description	Expansion %		Difference %
	Average Measured	Calculated	
Unconfined, SP	0.818	0.333	59
1D-SP	0.818	0.289	65
2D-SP	0.705	0.541	23
2D-TX	0.814	0.431	47

**Table 6 materials-17-00813-t006:** Results from *t*-tests of comparisons between digital and manual CI methods.

Horizontal Direction	Manual	Digital	Vertical Direction	Manual	Digital
Mean (mm/m)	3.55	3.90	Mean (mm/m)	1.25	1.29
Variance ((mm/m)^2^)	3.01	3.50	Variance ((mm/m)^2^)	0.56	0.56
Observations	24.00	24.00	Observations	24.00	24.00
Pearson Correlation	0.92		Pearson Correlation	0.91	
Hypothesized Mean Difference	0.00		Hypothesized Mean Difference	0.00	
df	23.00		df	23.00	
t Stat	−2.30		t Stat	−0.67	
*p* (T ≤ t) two-tail	0.05		*p* (T ≤ t) two-tail	0.05	
t Critical two-tail	2.07		t Critical two-tail	2.07	

**Table 7 materials-17-00813-t007:** The *t*-test results for comparisons between predicted CI and manual CI results.

Horizontal Direction	Predicted	Manual	Vertical Direction	Predicted	Manual
Mean (mm/m)	3.90	3.90	Mean (mm/m)	1.24	1.23
Variance ((mm/m)^2^)	3.50	2.24	Variance ((mm/m)^2^)	0.50	0.20
Observations	24.00	24.00	Observations	24.00	24.00
Pearson Correlation	0.80		Pearson Correlation	0.74	
Hypothesized Mean Difference	0.00		Hypothesized Mean Difference	0.00	
df	23.00		df	23.00	
t Stat	0.00		t Stat	0.07	
*p* (T ≤ t) two-tail	0.96		*p* (T ≤ t) two-tail	0.94	
t Critical two-tail	2.07		t Critical two-tail	2.07	

## Data Availability

Data are contained within the article.

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
