# Peer review of "An Integrated Framework for Image Acquisition, Processing, and Analysis Procedures for Automated Damage Evaluation of Concrete Surfaces"

_materials, 2024, doi:10.3390/ma17040813_

Round 1

Reviewer 1 Report

Comments and Suggestions for Authors

The manuscript titled " An integrated framework for image acquisition, processing, and analysis procedure for automated damage evaluation of concrete surface" would be an interesting fit for publication in Materials but the article would need revision for the following comments before publication.
1. How does the study's approach to integrating AI with image analysis align with or contribute to existing literature on AI applications in concrete condition assessment or related fields?
2. Considering the importance of controlled variables, how does the study address or discuss the standardization of data collection procedures, and how might this contribute to the broader conversation on data quality in concrete assessment?
3. Given the shift towards digitalization in visual inspections, what practical considerations or challenges might arise in implementing the proposed protocol in real-world scenarios? Are there potential limitations discussed in the literature that need attention?
4. How does the digital cracking index (CI) compare with traditional visual inspection methods in terms of accuracy and efficiency? Are there studies in the literature that provide insights into the practical advantages or challenges of such digital techniques?
5. Please highlight the potential transferability of the developed protocol beyond the specific case of ASR-induced cracking. Discuss how this protocol can be adapted for different types of concrete damage or in diverse environmental conditions.
6. Considering the linear trend observed in the expansion as a function of the CI, what are the potential future directions for research, and how might these align with or build upon existing literature on concrete cracking prediction and analysis?

Comments on the Quality of English Language

English language used is of high quality, please perform minor editing like grammar etc

Author Response

The authors would like to thank the reviewer for their time and contribution to the publishing process of this research work. All comments of improvements were addressed and highlighted in the manuscript. Please find the comments from the corrections to the manuscript below:

Review comment

Addressed by contributing author

The manuscript titled " An integrated framework for image acquisition, processing, and analysis procedure for automated damage evaluation of concrete surface" would be an interesting fit for publication in Materials but the article would need revision for the following comments before publication.

1. How does the study's approach to integrating AI with image analysis align with or contribute to existing literature on AI applications in concrete condition assessment or related fields?

A line was added in the introduction for clarity.

2. Considering the importance of controlled variables, how does the study address or discuss the standardization of data collection procedures, and how might this contribute to the broader conversation on data quality in concrete assessment?

Bullet point 2 in the conclusion was modified to include this as well as the scope of the work.

3. Given the shift towards digitalization in visual inspections, what practical considerations or challenges might arise in implementing the proposed protocol in real-world scenarios? Are there potential limitations discussed in the literature that need attention?

Thank you for this comment, it is one of the next steps of this research project to transfer the protocol to field structures. An additional bullet point in the conclusion was added to address this comment as well as one from another reviewer.

4. How does the digital cracking index (CI) compare with traditional visual inspection methods in terms of accuracy and efficiency? Are there studies in the literature that provide insights into the practical advantages or challenges of such digital techniques?

The accuracy and efficiency will depend on the quality of the images which would require a parametric study to find the accuracy level based on the parameters modified. Until this is performed, measuring overall accuracy can’t be generalized.

A line with a reference to a study describing thoroughly the pros and cons was added to the introduction.

5. Please highlight the potential transferability of the developed protocol beyond the specific case of ASR-induced cracking. Discuss how this protocol can be adapted for different types of concrete damage or in diverse environmental conditions.

This was addressed in the same line as for the second comment of item 4.

6. Considering the linear trend observed in the expansion as a function of the CI, what are the potential future directions for research, and how might these align with or build upon existing literature on concrete cracking prediction and analysis?

The last bullet point in the updated conclusion reflects this comment. The intention is to then understand the relationship between internal damage and external response.

Reviewer 2 Report

Comments and Suggestions for Authors

The paper explores a framework designed for machine-assisted analysis of cracks on concrete surfaces. This research aims to utilize AI-integrated image analysis to develop an improved, quantitative method for evaluating concrete damage, advancing beyond traditional qualitative approaches. Assessing the condition of concrete involves crucial observation, especially regarding non-structural cracks. Human observations, though, are both subjective and laborious. This study holds significance due to the rise of artificial intelligence, which can empower machines to visually interpret concrete surface images, providing valuable insights into crack severity. The adoption of digital visual inspections is becoming standard practice, notably in tasks like assessing concrete surface cracking. Alkali-silica reaction (ASR) is one mechanism causing distinct map-like cracking, quantified through the cracking index (CI) to gauge deterioration levels. I believe the key contributions of the study are the developments in analyzing the complex cracking caused by detrimental ASR phenomena.

I suggest authors rework figures captions, e.g., line 137., “Figure Error! No text of specified style in document.. (a) ..” The word “Error!” and  “..” should be fixed.

The manuscript contains numerous images, yet their sources lack clarity, prompting the need for authors to furnish additional details about their origins.

The authors have provided a comparison between manual and digital evaluations. However, I believe that the manuscript would greatly benefit from a validation study comparing its findings to the existing literature on the traditional assessment methods of ASR cracking. This comparative analysis would enhance the manuscript's credibility and contribute significantly to its overall strength.

I am intrigued by the methodology presented in this study and would appreciate further insight into its performance under real-world scenarios. Concrete elements often exhibit a combination of mechanical and chemical cracking on their surfaces in practical conditions. Therefore, delving deeper into how the proposed methodology addresses and distinguishes between these two types of cracks in such complex settings would significantly enhance the applicability and reliability of this approach. Discussing the methodology's efficacy in accurately identifying and differentiating between mechanical and chemical cracking within concrete structures will not only strengthen its practicality but also provide invaluable insights for real-world implementation.

I think the references look good to me.

Author Response

The authors would like to thank the reviewer for their time and contribution to the publishing process of this research work. All comments of improvements were addressed and highlighted in the manuscript. Please find the comments from the corrections to the manuscript below:

Review comment

Addressed by contributing author

The paper explores a framework designed for machine-assisted analysis of cracks on concrete surfaces. This research aims to utilize AI-integrated image analysis to develop an improved, quantitative method for evaluating concrete damage, advancing beyond traditional qualitative approaches. Assessing the condition of concrete involves crucial observation, especially regarding non-structural cracks. Human observations, though, are both subjective and laborious. This study holds significance due to the rise of artificial intelligence, which can empower machines to visually interpret concrete surface images, providing valuable insights into crack severity. The adoption of digital visual inspections is becoming standard practice, notably in tasks like assessing concrete surface cracking. Alkali-silica reaction (ASR) is one mechanism causing distinct map-like cracking, quantified through the cracking index (CI) to gauge deterioration levels. I believe the key contributions of the study are the developments in analyzing the complex cracking caused by detrimental ASR phenomena.

Thank you immensely for taking the time to review this paper. You have described the goal precisely which validates our motivation for this work.

I suggest authors rework figures captions, e.g., line 137., “Figure Error! No text of specified style in document.. (a) ..” The word “Error!” and  “..” should be fixed.

This was addressed.

The manuscript contains numerous images, yet their sources lack clarity, prompting the need for authors to furnish additional details about their origins.

All of the images are original to this work and taken by the authors for the purpose of this work only.

The authors have provided a comparison between manual and digital evaluations. However, I believe that the manuscript would greatly benefit from a validation study comparing its findings to the existing literature on the traditional assessment methods of ASR cracking. This comparative analysis would enhance the manuscript's credibility and contribute significantly to its overall strength.

Thank you for this constructive comment. As per the current practice in assessing ASR damage (Ref. 4), the cracking index manually recorded along with measurements of volumetric expansion from Demec points installed on the structure is the conventional method. Fortunetaly in this study, we recorded the expansions (Table 1 and 5) and calculated them from the cracking index data (Eq. 2). This equation is an improved version of what has been previously developed for unreinforced blocks. I have added a paragraph in the last part of the discussion to bring forward this statement and the need to refine the models that were developed.

I am intrigued by the methodology presented in this study and would appreciate further insight into its performance under real-world scenarios. Concrete elements often exhibit a combination of mechanical and chemical cracking on their surfaces in practical conditions. Therefore, delving deeper into how the proposed methodology addresses and distinguishes between these two types of cracks in such complex settings would significantly enhance the applicability and reliability of this approach. Discussing the methodology's efficacy in accurately identifying and differentiating between mechanical and chemical cracking within concrete structures will not only strengthen its practicality but also provide invaluable insights for real-world implementation.

Thank you kindly for this comment, this is the intention of the research as a next phase. Access to real structures affected by ASR presenting known levels of damage is difficult however, we are currently developing another study to apply the methodology of this current work to real structures. Moreover, in this current study, we could isolate ASR cracking which will help us in the future when other crack types are involved. I have added a bullet point to the conclusion to highlight this.

Reviewer 3 Report

Comments and Suggestions for Authors

Author Response

Review comment

Addressed by contributing author

 The manuscript is generally clearly written and well-organized. It is a good fit for the journal. My comments are all related to minor presentation issues

Thank you for your comment, it is greatly appreciated.

 Abstract: The present version focuses on the aims of the study, without mentioning to results of the research. This information needs to be added.

Details of the outcomes of the work were added to the abstract and highlighted in yellow in the revised version of the manuscript.

 Line 11: “Visual inspection, a commonly used and versatile concrete condition assessment technique, is employed to assess concrete degradation by visually detecting signs of damage on the surface level…”

Comment: The wording is redundant. I suggest: “Visual inspection, a commonly used and versatile concrete condition assessment technique, is employed to assess concrete degradation by observing signs of damage on the surface level….”

Addressed.

 Figure 1 caption: “Concrete column affected by ASR presenting a map-cracking pattern”

Comment: I’m not sure about the wording, the present form seems to imply that ASR cracks maps.

This terminology is used in the ACI 201.1R:

2.1.2.7 Map cracking—1) intersecting cracks that

extend below the surface of hardened concrete; caused by

shrinkage of the drying surface concrete that is restrained by

concrete at greater depths where either little or no shrinkage

occurs; vary in width from fine and barely visible to open

and well-defined; or 2) the chief symptom of a chemical

reaction between alkalis in cement and mineral constituents

in aggregate within hardened concrete; due to differential rate of volume change in different members of the concrete;

cracking is usually random and on a fairly large scale and, in

severe instances, the cracks may reach a width of 12.7 mm

(0.50 in.) (Fig. 2.1.2.7(a) and (b)). (See also checking and

crazing; also known as pattern cracking.)

And sometimes used hyphenated such as in Ref. 4. I have added the ACI reference into the text.

 Line 127: As shown in Figure 2a, each CI reference grid should contain a “measurable amount of cracking”

Comment: What is the purpose of the quotation marks? Is this a quotation from a reference?

Quotation marks removed.

 Line 137: “Figure Error! No text of specified style in document..”

Addressed.

 Line 158: Commercial software compatible with the camera facilitates the connection of the camera to a computer for remote shooting and camera control via a USB cable or Wi-Fi connection”

Comment: What commercial software was used? Is this software that came with the camera, or an aftermarket product?

Commercial software that came with the camera, Canon in this case (https://www.canon.ca/en/Features/EOS-Webcam-Utility). The authors did not want to add the specific name to the manuscript since we want to promote accessibility and not limit the work to one type of camera. A line was added to clarify that the software came with the camera.

 Line 164: “Compact and portable LED light kits (containing 480 LED units) were used to enhance the visibility of crack features”

Comment: How were three different color temperatures (white, warm white, yellow) obtained from the LED’s ? Product information needs to be provided.

The LEDs colours or white balance are modifiable from the unit itself which is common for many lights available on the market (https://neewer.com/collections/bi-color-led-lights/products/nl480-led-panel-lights-66600538). A line was added to describe the specifications of the lights to ensure repeatability. Thank you for this comment.

 Figure 10: It seems odd to me that the color temperature of the light is depicted in the graphs as unrepresentative colors, e.g., white-yellow is depicted with reddish orange color. Also, the caption should be modified to explain that the color names refer to the color balance of the illumination, not the color of the concrete.

Modifications to the bars were made to match the description of the colour balance. The caption was also modified as per the comment.

For Figures 11-16, CI values are shown to have units of mm/m, These units are missing for the CI values listed in Tables 6 & 7, and in Figure 15.

This was corrected.

The figures skip from Figure 17 to Figure 25. Figure 25 presumable should be listed as Figure 18, and it needs to be cited in the text.

This was corrected and text modified.

Figure 25: Why are horizontal residuals shown as scatter diagrams, and vertical residuals depicted as a bar chart?

The figure on the left represents the residuals plot and the figure on the right is the corresponding histogram to determine the normality. The figure labels and caption were corrected for clarity.  

Line 576: Digitalization of visual inspections is becoming the standard practice

Comment: I confess to being “old school”, having begun my scientific use of computers in the era when we used time-shared mainframe computers with programs written as punched cards, but I prefer the term “digitizing” rather than “digitalizing”.

Corrections were made to the text.